# Phase Compensation of the Non-Uniformity of the Liquid Crystal on Silicon Spatial Light Modulator at Pixel Level

**DOI:** 10.3390/s21030967

**Published:** 2021-02-01

**Authors:** Zhen Zeng, Zexiao Li, Fengzhou Fang, Xiaodong Zhang

**Affiliations:** State Key Laboratory of Precision Measuring Technology & Instruments, Laboratory of Micro/Nano Manufacturing Technology, Tianjin University, Tianjin 300072, China; zengzhen@tju.edu.cn (Z.Z.); zexiaoli@tju.edu.cn (Z.L.); fzfang@tju.edu.cn (F.F.)

**Keywords:** spatial light modulator, phase compensation, optical interferometry, holographic optical element

## Abstract

Phase compensation is a critical step for the optical measuring system using spatial light modulator (SLM). The wavefront distortion from SLM is mainly caused by the phase modulation non-linearity and non-uniformity of SLM’s physical structure and environmental conditions. A phase modulation characteristic calibration and compensation method for liquid crystal on silicon spatial light modulator (LCoS-SLM) with a Twyman-Green interferometer is illustrated in this study. A method using two sequences of phase maps is proposed to calibrate the non-uniformity character over the whole aperture of LCoS-SLM at pixel level. A phase compensation matrix is calculated to correct the actual phase modulation of the LCoS-SLM and ensure that the designed wavefront could be achieved. Compared with previously known compensation methods, the proposed method could obtain the phase modulation characteristic curve of each pixel on the LCoS-SLM, rather than a mono look-up table (LUT) curve or multi-LUT curves corresponding to an array of blocks over the whole aperture of the LCoS-SLM. The experiment results show that the phase compensation precision could reach a peak-valley value of 0.061λ in wavefront and this method can be applied in generating freeform wave front for precise optical performance.

## 1. Introduction

The development of Micro-nano manufacturing techniques [1,2,3] brings great improvements to the spatial light modulator (SLM) and makes it possible to be employed in the advanced optical applications. The active research fields of SLM include adaptive optics [4], optical vortex [5], diffractive optical elements (DOEs) [6,7] and holographic displays [8]. With a function to generate some equally spaced phase constants, the liquid crystal on silicon spatial light modulator (LCoS-SLM) can be used as a phase shifter in the measurement without a mechanical scanning such as a Piezoelectric Transducer (PZT), especially in some microscope systems to measure Micro-electromechanical Systems (MEMS) elements [9] or microstructures [10]. The LCoS-SLM also shows the ability to generate designed wavefront, which can be used to provide the flexibility necessary to change the overall system focal length [11].

The LCoS-SLM is hopefully to be considered as an ideal optical element to modify the complex amplitude of the incident wave by a simple multiplication. However, the phase modulation characteristic of LCoS-SLM is not perfectly linear, which is limited by its physical structure and working principle. There is a complicated internal behavior in the actual sandwich-like thin layers of the LCoS-SLM when it works. Many researches on the nonlinear characteristics of liquid crystal modulation have been reported. Among them, the crossed-polarizers (C-P) method [12] and the diffraction-based calibration method [13], Michelson interferometer method [14,15] are used to measure the phase response of the LCoS-SLM. Furthermore, a self-generated grating diffraction method [16] is also recently reported.

A common idea to solve the problem is to build a mono global look-up table (LUT) which expresses the relationship of greylevel of input grey-level map and the phase retardation of the output wavefront and apply the LUT when addressing a grey-scale map. In transmission and reflective mode LCoS-SLM applications, researchers found that there were phase non-uniformity phenomenon despite the global LUT method has been used, which caused some bad optical performance such as fuzzy star image in adaptive optics or low-quality diffraction imaging result. Thus, the measurement and calibration to the phase modulation nonlinearity and non-uniformity of LCoS-SLM becomes the basis of relative optical research application. 

An important improvement to solve this problem is the Multi-LUT method, which is proposed by Oton [17]. The key of this method is to separate the sub-apertures of the LCoS-SLM and measure the phase response for each cell. Ronchi grating method or Michelson interferometric method is used to measure the phase response and build a Multi-LUT. However, calibrating each cell of the defined grid with the input greylevel from 0 to 255 is time-consuming work. Lu presents a method using a special gray pattern to shorten the phase measurement process [18] with the help of a Fizeau interferometer. Zhang’ work [19] presents a node interpolation method to speed up the calculation process. On the other hand, the inherent problem of Multi-LUT method cannot be avoid. One is that the number and size of sub-aperture is limited by the possible captured pixels by the camera for each cell [17] or the resolution limit of the Ronchi grating method. The other is the high-order polynomials fitting errors introduced, which is significant at the edge part of the LCoS-SLM [20,21]. For every sub-aperture, a fitting by high-order polynomials is employed to cover the cell and get the Multi-LUT table to compensate the non-uniformity phase of the LCoS-SLM. The discontinuity between different cell could be improved in theory if the number of the cells increase, which means a global pixel-to-pixel compensation. Besides the Multi-LUT method, other researchers implement iterative compensation method to compensate the initial wavefront of LCOS-SLM to a nearly plane wavefront, with a λ/16 PV and an uncertainty of λ/33 reported [22]. He presented a wavefront correction for spatial nonuniformity of the liquid crystal on silicon based spatial light modulator and points out that the phase-shift method is the best phase retrieval techniques to calculate the nonuniformity map of the LCoS backplane [23]. Xia presented a Pixel-addressable phase calibration of spatial light modulators using a common-path phase-shifting interferometric microscopy approach [24].

In this paper, we present an interferometric based method to analyze these causes of phase non-linear and non-uniform distortion of the LCoS-SLM theoretically and experimentally. By addressing two sequences of special gray pattern, the pixel-to-pixel compensation of LCoS-SLM wavefront distortion can be achieved. The convenience of the phase-shift interferometer to measure the instantaneous phase could be easily applied and then the phase compensation look-up matrix could be computed as well. The method avoids the errors in sub-aperture measurement and high-order polynomials fitting. It can build a close-loop of phase modulation and measurement and ensure the LCoS-SLM obtain the ability to generate accurate phase and wavefront as desired.

## 2. Phase Modulation Non-Uniformity Analysis of LCoS-SLM

The phase modulation theory of LCoS-SLM must be illustrated at first. The structure of the LCoS-SLM is a sandwich-like layers, filled with parallel nematic liquid crystal. It has arrayed pixel units structure, each of which is under the control of a square-wave voltage, driving the liquid crystal molecules to rotate their director orientation. When passing the liquid crystal molecules, the extraordinary light index would change according to the director orientation so that the optical path and the phase of the light of different pixel unit is changed. It is called electronically controlled birefringence effect. For a reflective LCoS-SLM, the phase change [17] is:
(1)$$\Delta \phi (x,y,N)=\frac{2\pi }{\lambda }(n_{e}(x,y,N)-n_{o})d(x,y)$$
in which, *n_o_* is the index of ordinary light and *n_e_* is the index of extraordinary light, *d* is double thickness of the liquid crystal layer and *N* is the designed greylevel. And the light waves travel in a round trip in the liquid crystal layer, which means the optical path difference (OPD) and phase retardation would be doubled.

Theoretically, the wavefront generated by a reflecting surface can be affected either by extraordinary light index changes or by profile changes. The LCoS-SLM studied in the paper is a phase-only SLM using kinoform to change the incident light. The kinoform is a phase hologram in which the phase modulation is introduced by a surface relief profile. The phase modulation principle of the LCoS-SLM is that when addressing a gray image with a grayscale value, the LCoS-SLM can generate a phase retardation distribution to the incident light by the literature [17]
(2)$$\Delta \phi (x,y,N)=\Delta \phi_{Norm}(N)\frac{2\pi }{\Delta \phi_{Norm}[N_{2\pi }(x,y)]}$$
where *N* is the gray level value, *N*_2π_(*x*, *y*) is the gray level corresponding to a 2*π* phase modulation in the position (*x*, *y*) and Δ*ϕ*(*x*,*y*,*N*) is the phase retardation distribution. Thus, with a global *LUT* table, the LCoS-SLM is then addressed according to the kinoform calculated by
(3)$$\Phi_{kinoform}(x,y,\Delta \phi_{desired}(x,y))=LUT(\Delta \phi_{desired}(x,y))$$
where Φ*_kinoform_* is the actual kinoform with a *LUT* value of the desired phase at point (*x*, *y*) processed with the phase wrapping algorithm. It is then transferred to the gray level to be load on the LCoS-SLM.

According to the phase modulation principle of LCOS-SLM, its ability to change the phase rely on the rotation of the liquid crystal molecule. Previous studies reported about the reasons of phase distortion of the LCoS-SLM, where main contribution comes from the backplane curvature of the LCoS-SLM during its polishing and cutting process [25]. The coverglass on the top layer could introduce some wavefront error. The liquid crystal anchor direction is not ideal the same especially near the coverglass or the backplane so its effect on the incident ray is complicated. The control strategy of driven circuit of LCoS-SLM [20] could lead to different phase response. Moreover, the working temperature or local heat brought by laser beam would lead to the bend of the backplane as well as liquid crystal viscosity change, which reduces the phase stability [20]. 

The phase distortion of LCoS-SLM is a combined action by all above reasons, showing as a significant initial phase residual error and phase modulation non-linearity and non-uniformity. To solve it, setting lower temperature could reduce the phase fluctuation [26], however it would also change the phase response character of the LCoS-SLM. A measurement example of an inactive LCoS-SLM is shown in Figure 1a, with the environmental temperature of 22 °C. The result shows that the LCoS-SLM has an initial inherent wavefront distortion (WFD) rather than an ideal plane wave. Its Zernike polynomials fitting coefficients are shown in Figure 1b. The main contribution of the wavefront distortion comes from the low order aberration, especially the spherical aberration.

In short, for a LcoS-SLM device, its phase response must be re-calibrated at different working wavelength and temperature to compensate its phase non-linearity and non-uniformity. Any phase modulated by the LcoS-SLM would be affected by its phase modulation performance.

## 3. Phase Non-Uniformity Compensation Method

The strategy of the phase compensation method is shown in Figure 2. First of all, by addressing a gray pattern (kinoform) with zero grey value which means that the LcoS-SLM is inactive, the inherent WFD could be got by the interferometer using phase-shifting method. The gray level is set to 0. Then by addressing two special gray pattern sequences, two phase retardation non-uniformity characters (PRNU) could be measured. Two PRNUs are calculated to get the phase compensation matrix (*PCM*) by a merging algorithm. The matrix contains the compensation coefficient for each pixel of the LcoS-SLM. After that, the *PCM* could be used to compute the global pixel-to-pixel compensated phase for the inherent WFD. By phase inversion and superposition process, the inherent WFD is compensated to a plane wavefront. On the other hand, the *PCM* could be used to correct the superimposed phase calculated from the designed wavefront and the inverse inherent WFD to get a revised gray pattern, which could generate the desired wavefront when loaded on the LcoS-SLM.

For some experiments which only need to generate a plane wavefront, the inherent distorted wavefront of LcoS-SLM can be corrected by two methods. One is the iterative compensation, which need about three times measurement and calculation and could reduce the distortion to better than 1/10λ. The other is the phase non-uniformity compensation method, by measuring the inherent WFD once and applying the *PCM* to calculate the actual revised phase and gray pattern to load on the LcoS-SLM.

The operation of the phase non-uniformity compensation method is illustrated in detail as follows. After global phase non-linearity measurement and calibration, a global LUT could be obtained. Based on the interferometric system’s ability to measure each point’s relative phase response, a special phase map is designed as Figure 3.

The whole phase map is divided into three parts. The area outside the rectangle ABCD is set with the designed phase, changing from 0 to 2π. The area of inner rectangle EFGH is set with a reference phase of zero. The middle part between them is assigned with a gradual changing phase value to connect the designed phase the reference phase part. Then each pixel’s phase change is calculated by subtracting the reference phase in the inner rectangle. The interferometer could get the phase of the measured wavefront from optical path difference (OPD) data. It normally shows a result with the tilt section of the wavefront removed, which is not needed in the proposed method. It would also disturb the overall consistency of the measuring sequence. Hence, the tilt section should be reserved as the raw measured data and it could be adjusted by subtracting the first measured reference wavefront in the sequence. During the calibration process, the input map’s greylevel changes from 0 to 255 with a rectangular reference area set as 0. The data from the first series is the global phase response except the reference rectangle on the left. And then the second series of phase maps are used to get the opposite part of phase response. At last, the two part of data is combined to generate the global phase response of the LcoS-SLM by a merging algorithm. In Figure 4, both left and right sequences of 18 holograms to calibrate the phase response of the LcoS-SLM before and after inherent WFD compensation are presented. In this study, the direct measured OPD data of whole LcoS-SLM aperture is used for calculation and phase compensation.

From the global phase response, the pixel-to pixel phase linearity compensation coefficient can be obtained by a reciprocal calculation to get the phase compensation coefficient matrix, which is equivalent to a three-dimensional matrix. With the 3-D matrix, the phase response compensation for each pixel of the LCoS-SLM could be got.

Then considering the need to generate a designed phase or wavefront, the math model of the PRNU compensation method is illustrated as follows. The designed phase of the LcoS-SLM could be expressed as a matrix as
(4)$$\Phi_{design}=\left[\begin{array}{cccc} \varphi (x_{1},y_{1}) & \varphi (x_{2},y_{1}) & \cdots & \varphi (x_{m},y_{1})\\ \varphi (x_{1},y_{1}) & \varphi (x_{2},y_{2}) & \cdots & \varphi (x_{m},y_{2})\\ \cdots & \cdots & \cdots & \cdots \\ \varphi (x_{1},y_{n}) & \cdots & \cdots & \varphi (x_{m},y_{n}) \end{array}\right]$$
and phase response matrix (*PRM*) of the input grey pattern is
(5)$$PRM=\left[\begin{array}{cccc} G(1,1) & G(1,2) & \cdots & G(1,n)\\ G(2,1) & G(2,2) & \cdots & G(2,n)\\ \cdots & \cdots & \cdots & \cdots \\ G(m,1) & \cdots & \cdots & G(m,n) \end{array}\right]$$
(6)$$G(m,n)={\left[\begin{array}{cccc} g_{0} & g_{1} & \cdots & g_{255} \end{array}\right]}_{\left(m,n\right)}^{T}$$
in which *g* is the phase value from 0 to 2π.

Phase correction matrix (*PCM*) is calculated from the *PRM*’s data.
(7)$$PCM=\left[\begin{array}{cccc} C(1,1) & C(1,2) & \cdots & C(1,n)\\ C(2,1) & C(2,2) & \cdots & C(2,n)\\ \cdots & \cdots & \cdots & \cdots \\ C(m,1) & \cdots & \cdots & C(m,n) \end{array}\right]$$
(8)$$C(m,n)={\left[\begin{array}{cccc} \frac{F_{lut}(g_{0})}{g_{0}} & \frac{F_{lut}(g_{1})}{g_{1}} & \cdots & \frac{F_{lut}(g_{255})}{g_{255}} \end{array}\right]}_{\left(m,n\right)}^{T}$$
in which *C*(*m*,*n*) is calculated from the measured phase and the desired phase with a look-up-table function for each pixel as Formula (4).

Finally, the compensated phase map is generated from *GL_out_*, which is the gray level value of the output kinoform.
(9)$$GL_{out}=\Phi_{design}\cdot PCM=\left[\begin{array}{cccc} F(\varphi (x_{1},y_{1})) & F(\varphi (x_{2},y_{1})) & \cdots & F(\varphi (x_{m},y_{1}))\\ F(\varphi (x_{1},y_{1})) & F(\varphi (x_{2},y_{2})) & \cdots & F(\varphi (x_{m},y_{2}))\\ \cdots & \cdots & \cdots & \cdots \\ F(\varphi (x_{1},y_{n})) & \cdots & \cdots & F(\varphi (x_{m},y_{n})) \end{array}\right]$$
(10)$$F(\varphi (x_{m},y_{n}))=\varphi (x_{m},y_{n})/2\pi \times 255\cdot C(m,n)$$

For each pixel the designed phase is corrected with the *PCM*. Through the whole process, the phase shift for each gray level addressed to each pixel unit on the LcoS-SLM can then be precisely controlled and implemented to generate the desired wavefront.

## 4. Experiments and Results

### 4.1. Optical Setup

The optical setup is illustrated in Figure 5. The LcoS device used in the experiment is fabricated by HOLOEYE Photonics AG Company (Berlin, Germany), having 1920 × 1080 pixels with a pixel pitch of 8 μm × 8 μm and 87% fill factor. Its nominal size is 15.36 mm × 8.64 mm and the zero order diffraction efficiency is 60%. In the system, a Twyman-Green interferometer is used with the wavelength of 632.8 nm. It is calibrated with a precision optical flat of 1/20λ. Its output light is linearly polarized with an aperture of 50.8 mm, so a mounted polarized beam splitter (PBS) cube and a charge-coupled device (CCD) camera is used to adjust the polarization direction shown in Figure 6. The direction of red arrow is the polarization direction of the laser light, and the blue one means that of the PBS. In Figure 6a, by adjusting the interferometry lens rotation angle and observing the CCD image and its peak shape, the well-matched polarization direction could be found out. In Figure 6b, the polarization directions are perpendicular and the grey histogram of the CCD image is lowest. As the PBS prism rotate with 90 degrees from along the polarized direction to the perpendicular direction, the image of the PBS prism goes darker and disappear. Thus, the grey level histogram in Figure 6a has two peaks and reduce to one in Figure 6b.

Then the LcoS-SLM replace the camera’s position and a spherical kinoform is imported on the LcoS-SLM to help adjusting the relative tilt angle between the interferometer’s plane wavefront and the LcoS-SLM head. The tilted and adjusted interference patterns are shown in Figure 6. Not all the areas of the captured image should be used to calculate the data for the phase or kinoform designed, so it is necessary to set a mask for the active area of the LcoS-SLM. A marked cross-shaped pattern is used to get the mask region, shown as Figure 7. It could adjust the orientation of the CCD image and the OPD data precisely and build a matching relationship. In the experiments, interferograms of 1920 × 1080 LcoS pixels are all imaged on 298 × 164 CCD pixels, and the data need to be adjusted according to the mask and the rotation angle of the polarized incident ray.

The total phase modulation depth of the LCoS-SLM is limited to 2π, for the benefit of phase-unwrapping algorithm used in the interferometer software. Therefore, the LcoS-SLM has a 2π phase encoded in 146 levels with a “5-5” electrical addressing sequence configuration, which is shown in Figure 8a. The sequence “5-5” means the duty cycle of the LcoS-SLM driving voltage, which makes the liquid crystal molecules tilted to modify the phase of the extraordinary light. Typical flicker of 5-5 configuration is shown in Figure 8b.

After these preprocessing steps, the LCoS-SLM could be applied with the Twyman-Green interferometer to generate the test kinoform patterns and get the phase response for each gray level. The schematic diagram of the experiment is shown as Figure 9. The details of it are discussed in the following sections.

### 4.2. Phase Non-Uniformity Measurement

Following the adjustment method and preparation work, the system is appropriate for the phase response measurement of the LCoS-SLM. As the proposed method, two sequences of designed phase patterns with the greylevel from 0 to 255 are imported to the LCoS-SLM to examine the phase non-uniformity performance.

To determine a unified reference phase value, the LCoS-SLM’s initial phase error is compensated by the fast iterative compensation method. The unified reference phase value is calculated by averaging the phase value within the reference area. From the result wavefront in Figure 10, nine point’s phase response curves are plotted in Figure 11, in which the non-uniformity is easy to identify.

It is important to note that the phase response on different pixel of the LCoS-SLM shows non-uniformity regardless of the inherent WFD of the LCoS-SLM is compensated or not. For better visualization of the interferograms, in this paper, the inherent WFD is compensated by the iterative compensation method in advance. The Figure 11 shows that at the edge where has bigger curvature of the backplane, the phase response is significantly different with that in the central part. The result means that the LCoS-SLM could not give a linear and uniform phase as the designed output requirement without the compensation, even after the initial phase distortion compensated. This character would reflect on every pixel of the LCoS-SLM on the final output phase.

### 4.3. Phase Non-Uniformity Compensation

Figure 12 shows the designed phase and revised phase with *PCM* compensation, as well as one sequence phase calculation results with a grey level step of 15 from 0 to 255 with the designed phase region on the right as an example. Each of the phase measurement results have been subtracted with the first measured phase to remove the tilt and background noise, so that the first presented calculation result shows as zero (the first green plane in Figure 12). After two PRNUs of LCOS-SLM are measured, the merging algorithm is used to get the phase compensation matrix. The wrapped inherent phase of LcoS-SLM and the revised phase with the *PCM* correction are presented in Figure 13. The phase texture distribution in the blue central part and the position of phase jumps at edge part are marked in these regions. To judge the effect of the proposed method, its compensation phase of Figure 12 is compared with the compensation phase of a three-time iterative compensation method, which could be employed to achieve a plane phase result of 0.079λ (Figure 14). The compensation phase difference calculated as a *P*-*V* value about 1/10λ, shown in Figure 15. Considering the wavefront residuals near the phase jump regions, the result means that the proposed phase method could achieve a precise control wavefront ability. The precision of it is at the same level compared with multiple interactive phase compensation method.

### 4.4. Method Verification

The verification of the proposed phase compensation method could be estimated by the optical performance and its wavefront quality. For some optical applications, the requirement to the output wavefront is to make it an approximate plane wavefront. For other advanced studies, especially that have high phase modulation precision requirement, the phase of output wavefront of LCoS-SLM needs to be precisely controlled. Otherwise it may generate some bad disturbance to the result. Visible proofs could be found from the interference patterns. For example, the bend interference patterns of plane wavefront change to straight and equal spacing straight interference patterns after phase compensation (Figure 16a,b). And it should be noticed that the shape of the measured wavefront with different compensation methods has obvious difference. With the mono LUT compensation, the residual wavefront shows as a spherical shape wavefront (Figure 17a), compared with a relative flat wavefront after the *PCM* compensation (Figure 17b). The wavefront aberration reduces from 0.188λ to 0.061λ, which is better than 1/16λ. The *PCM* compensation result is also better than the three-time iterative compensation method result (0.079λ) described in Section 4.3.

For more accurate requirement, the precise wavefront control could also benefit from the *PCM* compensation method. A freeform wavefront combined by a spherical wavefront and dual-sine wavefront is designed and generated to verify the precision of the wavefront generation ability of the LCoS-SLM with the *PCM* compensation, shown in Figure 16b. The formula of the freeform surface is expressed as
(11)$$Z=R-\sqrt{R^{2}-x^{2}-y^{2}}+0.1\times \sin((x/150)\times \pi )\times \sin((y/150)\times \pi )$$
where *R* is 20,000. The interference pattern of the phase is shown in Figure 18a. The measured optical path difference (OPD) data is present in Figure 18b and the wavefront aberration (Figure 18c) is 0.137λ (better than 1/7λ). Its shape is similar to a dual-sine wave with slight fluctuations less than 1/10λ.

The experiment results show that with the proposed method, the phase of LCoS-SLM could be precisely controlled pixel by pixel. The phase response non-uniformity of the LCoS-SLM has been compensated with the *PCM* and the LCoS-SLM could generated the phase or wavefront as designed. Figure 18c is the wavefront loaded with the corrected phase of a freeform phase, its wavefront errors are still retained at the 2π phase jump position and the edge with high slope. The error increases as the slope increases. In the example in our manuscript, the residual error of the freeform wavefront is at the level of 1/10λ. This result is lower than that of the calibration, but it is acceptable in optical applications.

The whole measurement and computation procedure could take about 13 min. The first part time consumption is to upload the two sequences of calibration hologram on the LCoS-SLM and get the phase data by the unwrapping algorithm of the interferometer. Because the software of the interferometer is not open source to us, so each phase map is got by manually operation, which takes about 10 s. The time cost of this part is 6 min. After these calibration phase data has been got, the rest computation to get the pixel-to-pixel *PRM* and *PCM* matrix could be done automatedly by a MATLAB program, which takes about 5 min. Then the hologram with *PCM* corrected would take about 90 s. As soon as it is uploaded on the LCoS-SLM, a new wavefront could be got. The experiment computing platform is a laptop with i7 CPU and 8 G memory.

It is important to note that the compensation is not a permanent affair, because the experiment environment changes every time. It is suggested to re-calibrate the LCoS-SLM with the proposed method periodically before the experiment.

## 5. Conclusions

A high accurate and effective method to calibrate the LCoS-SLM’s phase response is proposed. The calibration configuration consists of an interferometer and a LCoS-SLM to for the phase response measurement to get a global pixel-to-pixel phase compensation matrix. In the verification experiment, the wavefront aberration reduces from 0.188λ to 0.061λ, which is better than 1/16λ. It offers the optical measuring system using LCoS-SLM an accurate and stable wavefront generation ability. In the freeform wavefront generation experiment, a freeform wavefront combined by a spherical wavefront and dual-sine wavefront could be achieved with wavefront error less than 1/7λ. The application of LCoS-SLM would benefit with such phase compensation method in future research.

## Figures and Tables

**Figure 1 sensors-21-00967-f001:**
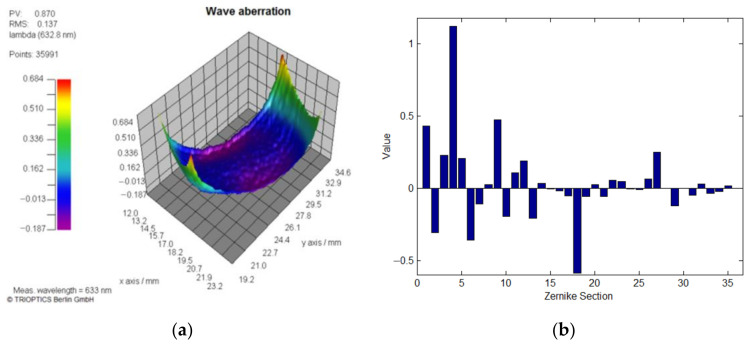
(**a**) The inherent WFD of LCoS-SLM and (**b**) its 36 Zernike coefficients sections.

**Figure 2 sensors-21-00967-f002:**
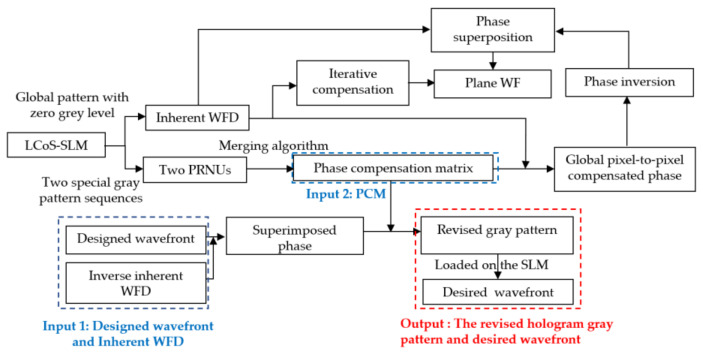
Strategy of phase compensation for LcoS-SLM.

**Figure 3 sensors-21-00967-f003:**
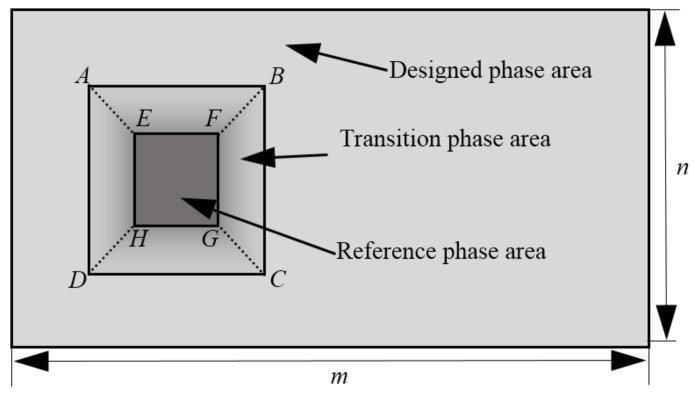
The gray map to get the global phase response of the LcoS-SLM.

**Figure 4 sensors-21-00967-f004:**
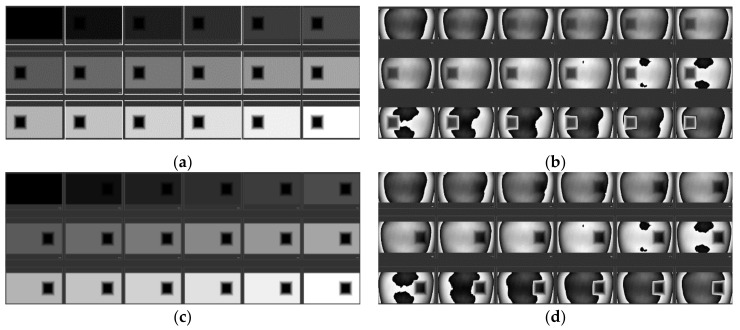
Left sequence of 18 phase response calibration holograms before (**a**) and after (**b**) inherent WFD compensation and right sequence of before (**c**) and after (**d**) inherent WFD compensation.

**Figure 5 sensors-21-00967-f005:**
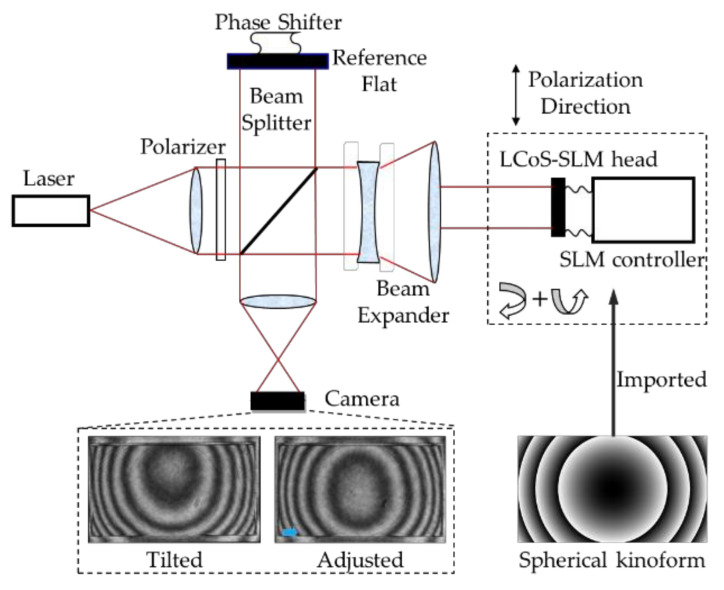
Optical setup and the adjustment procedure.

**Figure 6 sensors-21-00967-f006:**
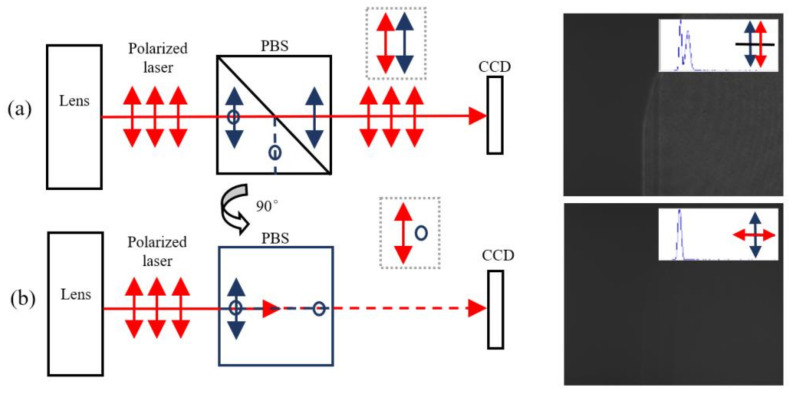
Polarization direction adjustment. (**a**) The PBS polarization direction is parallel with the laser light (**b**) The PBS polarization direction is perpendicular to the laser light.

**Figure 7 sensors-21-00967-f007:**
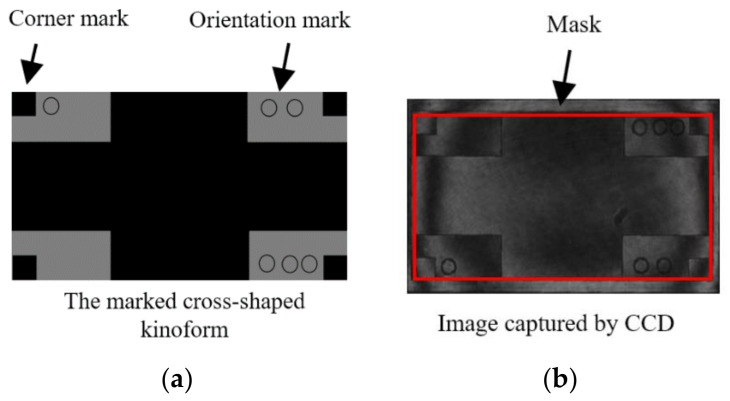
Mask matching procedure using a marked cross-shaped kinoform (**a**) to get the image (**b**).

**Figure 8 sensors-21-00967-f008:**
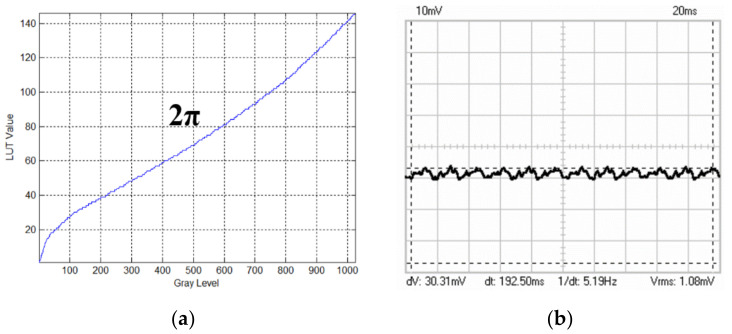
(**a**) LUT table encoded from 191 to 146 values to get 2π phase modulation. (**b**) Typical flicker of 5-5 configuration.

**Figure 9 sensors-21-00967-f009:**
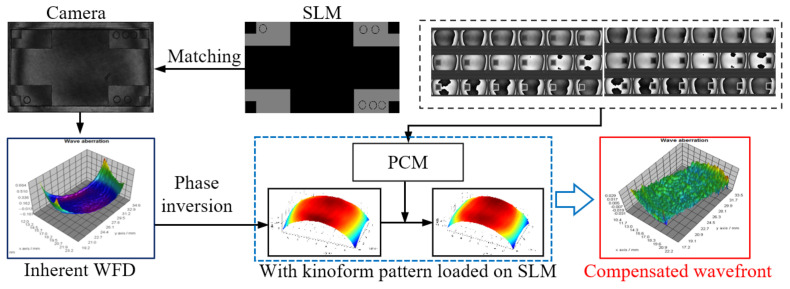
The schematic experimental diagram.

**Figure 10 sensors-21-00967-f010:**
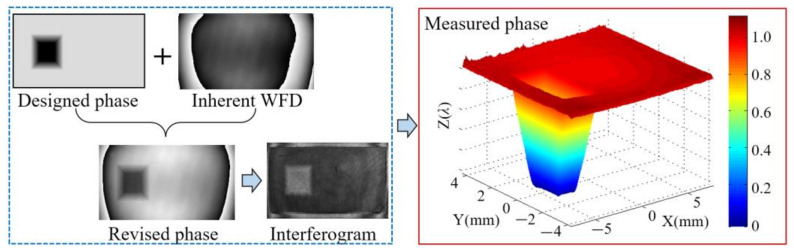
Process to get the phase retardation to different grey level.

**Figure 11 sensors-21-00967-f011:**
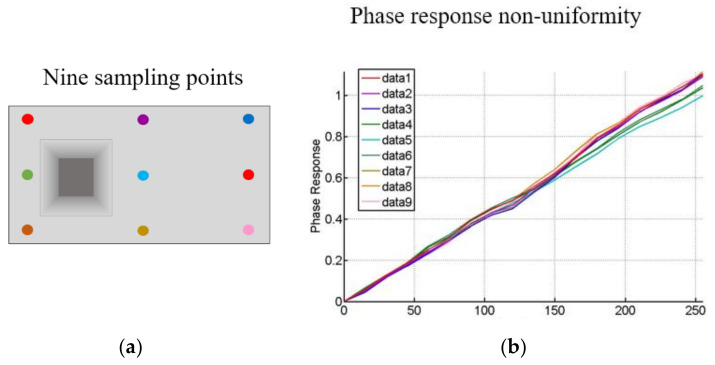
The phase non-uniformity performance of LCoS-SLM. (**a**) nine sampling points and (**b**) their phase response data to show the non-uniformity.

**Figure 12 sensors-21-00967-f012:**
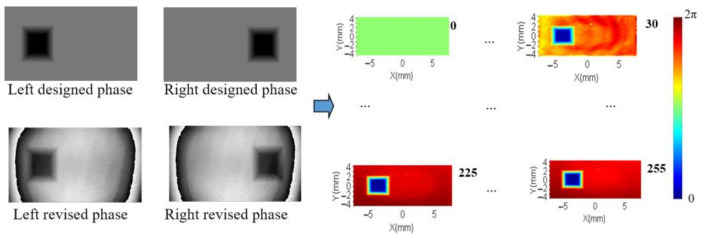
Process to get the global phase displacement with two series of phase maps.

**Figure 13 sensors-21-00967-f013:**
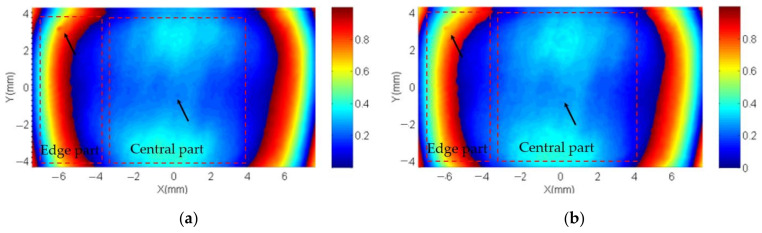
The wrapped inherent phase of LCoS-SLM. (**a**) The initial wrapped phase and (**b**) corrected wrapped.

**Figure 14 sensors-21-00967-f014:**
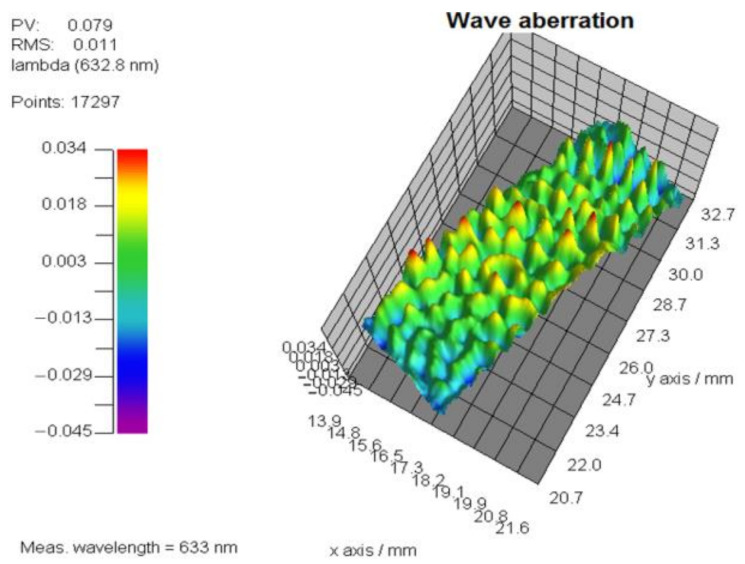
The iterative phase compensation result.

**Figure 15 sensors-21-00967-f015:**
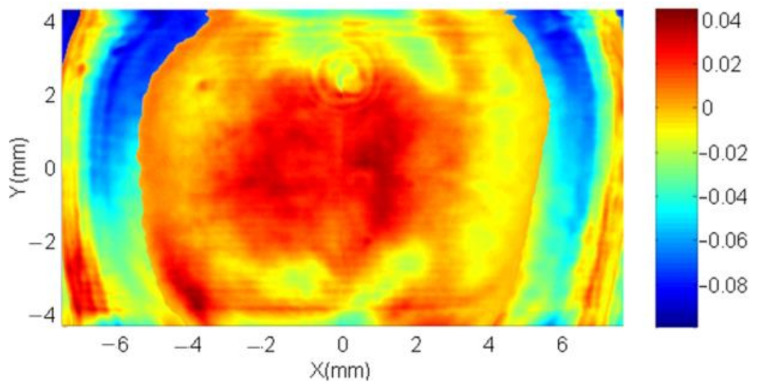
The compensation phase difference of the proposed method and the iterative phase compensation method.

**Figure 16 sensors-21-00967-f016:**
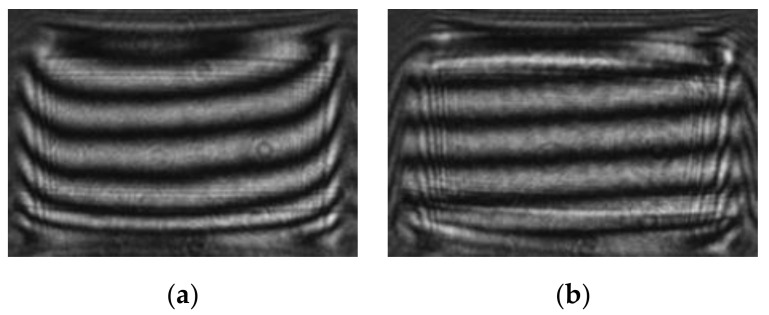
The interference fringes (**a**) before and (**b**) after phase compensation.

**Figure 17 sensors-21-00967-f017:**
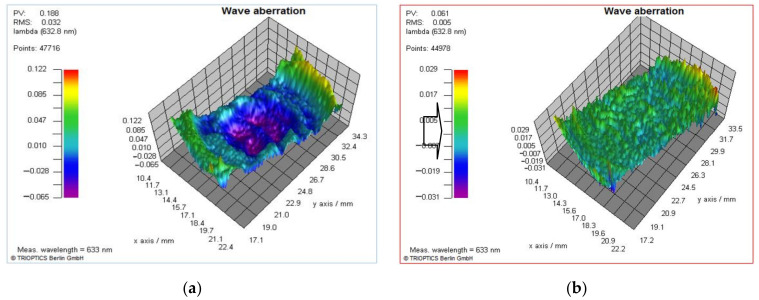
The measured wavefront with (**a**) mono-LUT method and (**b**) PCM method.

**Figure 18 sensors-21-00967-f018:**
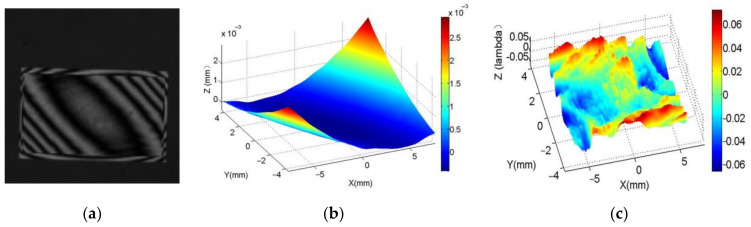
The freeform generated by LCoS-SLM and its deviation. (**a**) the interferogram (**b**) the OPD data (**c**) the wavefront aberration.

## Data Availability

The data presented in this study are available on request from the corresponding author.

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
