# Peer review of "Phase Compensation of the Non-Uniformity of the Liquid Crystal on Silicon Spatial Light Modulator at Pixel Level"

_sensors, 2021, doi:10.3390/s21030967_

Round 1

Reviewer 1 Report

This work proposes a phase-compensation method based on the calibration with two reference phase maps and further correction through lookup tables. The addressed topic fits the aims and scopes of Sensors. Moreover, the fundamentals and methodology are clearly described and the results are convincing. Even though the manuscript has its merits, some points must be improved:

  1. Sec. 2, line 92: change refractive indices ne and no to lower case. Variable G does not appear in equation (1);
  2. Sec. 3, Fig. 2: please highlight the input and output of the depicted algorithm by using arrows or different fill colors, for example;
  3. Sec. 4.1, Fig. 6: the interference patterns are too dark and hard to visualize;
  4. Sec. 4.1, Fig. 8: remove the subplot in the left side since it is superfluous;
  5. Sec. 4.2, Fig. 11: please use the same colors of the phase response plot to indicate the sampling points in (a) and include the x-axis label in (b);
  6. Sec. 4.3, Fig. 12: the surface plots in part (b) are illegible. Alternatively, it is possible to choose four gray level values and display the phase distribution as 2D plots;
  7. Please discuss the performance of your system regarding computational cost and processing time in comparison to the techniques proposed by other authors. Furthermore, it is mandatory to show how the average error and resolution are quantitatively improved in comparison to the state-of-the-art;
  8. As commented at the end of Sec. 4.4, it is necessary to perform periodic calibration due to the changes in the experimental conditions. In this sense, how stable and robust is your system (in quantitative terms) assuming operation in a temperature-controlled, vibration-free environment?

Author Response

Dear Editors and Reviewers:

We would like to express our sincere thanks for your constructive and positive comments on our manuscript. We have studied comments carefully and have made correction which we hope meet with approval. Revised portions are marked in red for easy tracking. The main corrections in the paper and the responds to the reviewer’s comments are as following:

Reviewer #1

Thank you for your careful review and comments in the manuscript. Firstly, thank you for your recognition of our research. We have provided more discussion in the introduction to enrich the background research. Then, based on the editor’s suggestion, we thought about it carefully and responded and revised them.

Comment.1: Sec. 2, line 92: change refractive indices ne and no to lower case. Variable G does not appear in equation (1);

Response: Thank you for your kind comments on the misnomer. We apologize for the improper expression. We have change the refractive indices to lower case. And the Variable G should be Variable N in equation (1).

Comment.2: Sec. 3, Fig. 2: please highlight the input and output of the depicted algorithm by using arrows or different fill colors, for example;

Response: Thank you for your careful work about this issue. More detailed descriptions are added to illustrate Fig.2. The input of the in the strategy of phase compensation for LCoS-SLM map should be the designed wavefront, the inherent WFD and the calculated PCM matrix. The output is the revised gray pattern of the compensated hologram to load on the LCoS-SLM.

Comment.3: Sec. 4.1, Fig. 6: the interference patterns are too dark and hard to visualize;

Response: Thank you for pointing out such problem. The CCD images in Fig.6 is the image of the mounted PBS prism in the optical configuration of Fig.6. As the PBS prism rotate with 90 degrees from along the polarized direction to the perpendicular direction, the image of the PBS prism goes darker and disappear. Thus, the grey level histogram in Fig.6(a) has two peaks and reduce to one in Fig6(b). We have modified both the text and the graphics to make them more visible.

Comment.4: Sec. 4.1, Fig. 8: remove the subplot in the left side since it is superfluous;

Response: Thank you for your careful work about this issue. We redraw the Fig.8 and removed the left subplot and we check relevant text description.

Comment.5: Sec. 4.2, Fig. 11: please use the same colors of the phase response plot to indicate the sampling points in (a) and include the x-axis label in (b);

Response: Thank you for your good suggestion on this problem. We modified the color of the sampling points of the phase data in Fig.11(a). We hope the correction would make it more proper to illustrate the different phase response with Mono LUT over the aperture of the LCoS-SLM.

Comment.6: Sec. 4.3, Fig. 12: the surface plots in part (b) are illegible. Alternatively, it is possible to choose four gray level values and display the phase distribution as 2D plots;

Response: Thank you for your kind advice. We redraw the Fig.12 and pick up four phase maps with greylevel of 0/30/225/255 as examples and show the figure as 2D plots.

Comment.7: Please discuss the performance of your system regarding computational cost and processing time in comparison to the techniques proposed by other authors. Furthermore, it is mandatory to show how the average error and resolution are quantitatively improved in comparison to the state-of-the-art;

Response: Thank you for your valuable comment. The whole measurement and computation procedure could take about 13 minutes. The first part time consumption is to upload the two sequences of calibration hologram on the LCoS-SLM and get the phase data by the unwrapping algorithm of the interferometer. Because the software of the interferometer is not open source to us, so each phase map is got by manually operation, which takes about 10 seconds. The time cost of this part is 6 minutes. After these calibration phase data has been got, the rest computation to get the pixel-to-pixel PRM and PCM matrix could be done automatedly by a MATLAB program, which takes about 5minutes. Then the hologram with PCM corrected would take about 90 seconds. As soon as it is uploaded on the LCoS-SLM, a new wavefront could be got. The experiment computing platform is a laptop with i7 CPU and 8G memory. We have added relevant descriptions to the manuscript.

And compared with the state-of-the-art researches, we use the interferometer to get the phase data of the whole aperture of the LCoS-SLM. The phase sampling points is 298 × 164 (with more well-match objective lens, the number could reach 800×500 with a 1K×1K interferometer camera), which is 750 times more than the 8 × 8 sampling grid method. That is the key point to get the pixel-to-pixel real phase data rather than calculating from the 8 × 8 grid sampling phase data. The process of experiment to get the pixel-to-pixel phase data is much easier and faster.

Comment.8: As commented at the end of Sec. 4.4, it is necessary to perform periodic calibration due to the changes in the experimental conditions. In this sense, how stable and robust is your system (in quantitative terms) assuming operation in a temperature-controlled, vibration-free environment?

Response: Thank you for your careful work about this issue. In our lab with a stable airflow, we used the environment test function of the interferometer software to measure the wavefront aberration within 10seconds. The test was executed several times during 20minutes and the largest wavefront distortion P-V value is about 1/20 λ. So, if there is no change of the environment such as air turbulence and temperature variation, the best of the reproducibility of the phase generated by the LCoS-SLM should be 1/20λ.

We appreciate for Editors and Reviewers’ warm work earnestly, and hope that the correction will meet with approval.

Once again, thank you very much for your comments and suggestions.

Best regards,

Corresponding authors:

Xiaodong Zhang, [email protected]

Reviewer 2 Report

Manuscript: crystals-1071817
Title: "Phase compensation of the non-uniformity of the liquid crystal on silicon spatial light modulator at pixel level"
Authors: Zhen Zeng, Zexiao li, Fengzhou Fang and Xiaodong Zhang

The manuscript proposes a phase modulation characteristic calibration and compensation method for liquid crystal on silicon spatial light modulator which surpasses previous methods in resolution and precision. It is claimed that the precision will allow the device to be used as a phase shifting element in classic interferometry.

The review of prior art is sufficient to explain the need for the present method, except for the omission of reference Jianpei Xia et al 2017 J. Opt. 19 125701. The manuscript can be improved by more clarity and exactness. For example, “It owes arrayed pixel units” is not clear and “No is the index of ordinary light”, where it should be no: the capital N refers to grey level, as described in eq. 2.

I have a number of questions and suggestions to improve the clarity of the manuscript:

  • Could the authors please provide references to substantiate the claim that “local heat brought by laser beam would lead to the bend of the backplane as well as liquid crystal viscosity change, which reduces the phase stability”?
  • The authors suggest that “by addressing a gray pattern (kinoform) with zero grey value which means that the LCoS-SLM is inactive”. Could the same effect be produced with the LCOS switched off?
  • What is the meaning of “zero order diffraction efficiency is 60%” in line 204?
  • I did not understand why only 298 × 164 CCD pixels are used to capture the 1920 × 1080 interferograms on line 222. Does this not reduce the resolution of the interferogram?
  • The authors state that “The sequence “5–5” means the duty cycle of the LCoS-SLM driving voltage” in line 229. Please illustrate this with the temporal waveform of the driving pulses for the “5–5” sequence.
  • In Fig. 11b the individual data lines should be cross-referenced to the points in Fig. 11a, so that the reader can see where is the data line for the central measurement point.
  • The right hand image in Fig. 12 and both images in Fig. 13 have no value because the former is too small and the latter shows no difference between left and right images. These figures should be omitted.
  • 17b shows a wavefront aberration of 0.061λ whereas the wavefront aberration in Fig.18(c) is 0.137λ. Presumably, the difference is because Fig. 17b is the wavefront from a device which is not driven whereas Fig.18(c) is a wavefront from a driven device. If this is correct, then the authors should add a sentence to explain this.

Correction and clarification of the above points will improve the manuscript.

Author Response

Dear Editors and Reviewers:

We would like to express our sincere thanks for your constructive and positive comments on our manuscript. We have studied comments carefully and have made correction which we hope meet with approval. Revised portions are marked in red for easy tracking. The main corrections in the paper and the responds to the reviewer’s comments are as following:

Reviewer #2

Thank you for your careful review and comments in the manuscript. Firstly, thank you for your recognition of our research. In view of the language problem that you pointed out, we have checked the writing grammars in detail throughout the article and revised them. Besides, we have provided more discussion in the introduction to enrich the background research. And we have added the reference (Jianpei Xia et al 2017 J. Opt. 19 125701) you suggested in the review of prior art. Then, based on the eight amendments you suggested, we thought about it carefully and responded and revised them.

Comment.1: Could the authors please provide references to substantiate the claim that “local heat brought by laser beam would lead to the bend of the backplane as well as liquid crystal viscosity change, which reduces the phase stability”?

Response: Thank you for your careful work on this issue. There should be a reference to illustrate the sentence. The information is Engström, D.; Persson, M.; Bengtsson, J.; Goksör, M. Calibration of spatial light modulators suffering from spatially varying phase response. Optics express 2013, 21, 16086.

Comment.2: The authors suggest that “by addressing a gray pattern (kinoform) with zero grey value which means that the LCoS-SLM is inactive”. Could the same effect be produced with the LCOS switched off?

Response: Thank you for your comments on the rigor of the article. It is different when addressing a gray pattern with zero grey value. The long axes of the liquid crystal molecules remain roughly in the same direction. However, when it is switched off, the liquid crystal molecules are in a free-floating state. When polarized light goes through these liquid crystal molecules, the change of its phase is not the same when it is addressed with zero value.

Comment.3: What is the meaning of “zero order diffraction efficiency is 60%” in line 204?

Response: Thank you for such valuable question. The zero order diffraction is an optical property of the hologram and such grating-like structure. The power of zero diffraction light take the largest part of the input light. In the experiment, we used the LCoS-SLM from HOLOEYE Company and its zero-order diffraction efficiency is 60%.

Comment.4: I did not understand why only 298 × 164 CCD pixels are used to capture the 1920 × 1080 interferograms on line 222. Does this not reduce the resolution of the interferogram?

Response: Thank you for your careful review and comments in the manuscript. The reason of the problem is because we used a two inches objective lens on the interferometer, which is restricted to the experimental equipment. Compared with the existing researches, we use the interferometer to get the phase data of the whole aperture of the LCoS-SLM. The phase sampling points is 298 × 164 (with more well-match objective lens, the number could reach 800×500 with a 1K×1K interferometer camera), which is 750 times more than the 8 × 8 sampling grid method. So the resolution of the captured interferogram is limited but it still could get more real phase data in a fast and convenient way than ever.

Comment.5: The authors state that “The sequence “5–5” means the duty cycle of the LCoS-SLM driving voltage” in line 229. Please illustrate this with the temporal waveform of the driving pulses for the “5–5” sequence.

Response: Thank you for your kind suggestion. We add the driving pulses of “5–5” sequence graph to illustrate this with the temporal waveform better. The graph is from the HOLOEYE LCoS-SLM document.

Comment.6: In Fig. 11b the individual data lines should be cross-referenced to the points in Fig. 11a, so that the reader can see where is the data line for the central measurement point.

Response: Thank you for pointing out our figure problem. We modified the color of the sampling points of the phase data in Fig.11(a). We hope the correction would make it more proper to illustrate the different phase response with Mono LUT over the aperture of the LCoS-SLM.

Comment.7: The right hand image in Fig. 12 and both images in Fig. 13 have no value because the former is too small and the latter shows no difference between left and right images. These figures should be omitted.

Response: Thank you for your valuable comment. We redraw the Fig.12 and pick up four phase maps with greylevel of 0/30/225/255 as examples and show the figure as 2D plots. And there are difference between Fig13(a) and (b), the phase texture distribution in the central blue part and the position of phase jumps should be clear and we mark these regions. Their compensation phase difference calculated is shown in Fig15.

Comment.8: 17b shows a wavefront aberration of 0.061λ whereas the wavefront aberration in Fig.18(c) is 0.137λ. Presumably, the difference is because Fig. 17b is the wavefront from a device which is not driven whereas Fig.18(c) is a wavefront from a driven device. If this is correct, then the authors should add a sentence to explain this.

Response: Thank you for your careful work on this issue. Fig. 17b is the plane wavefront measured by the interferometer with the compensated phase corrected by PCM. While Fig. 18c is the wavefront loaded with the corrected phase of a freeform phase, its wavefront errors are still retained at the 2π phase jump position and the edge with high slope. The error increases as the slope increases. In the example in our manuscript, the residual error of the freeform wavefront is at the level of 1/10λ. This result is lower than that of the calibration, but it is acceptable in optical applications.

We tried our best to improve the manuscript and made some changes in the manuscript. These changes will not influence the main content and framework of the paper and be marked in red in revised paper.

We appreciate for Editors and Reviewers’ warm work earnestly, and hope that the correction will meet with approval.

Once again, thank you very much for your comments and suggestions

Best regards,

Corresponding authors:

Xiaodong Zhang, [email protected]

Reviewer 3 Report

The manuscript develops novel methods to calibrate the phase response of a LCOS SLM. The method is presented with some details and it clearly shows high quality wavefront aberration compensation, accuracy of the order of 1/16 lambda. With the matrix approach, the phase shift for each grey level on the SLM is controled with high precision and is transfered onto each pixel pf the SLM. I note the following minor points to consider in the final form: 

  • The SLM and LC layer may introduce some noise due to scattered light, or diffraction by the electrode structures. Does it influences the precision of the aberration measurements ? 
  •  The strategy shown in Fig 2 requires computation time. Which is the time required to calculate and then to display the whole correction on each pixel of the SLM. At which speed to generate a new wave ? 
  • After correction and no change of the environment, which reproducibility of the X Y addressing electronics on the pixel phase ? 

  To conclude the manuscript is accepted for publication in the journal after including answers and comments to the questions.                                                                                                     

Author Response

Dear Editors and Reviewers:

We would like to express our sincere thanks for your constructive and positive comments on our manuscript. We have studied comments carefully and have made correction which we hope meet with approval. Revised portions are marked in red for easy tracking. The main corrections in the paper and the responds to the reviewer’s comments are as following:

Reviewer #3

Thank you for your careful review and comments in the manuscript. Firstly, thank you for your recognition of our research. In view of the language problem that you pointed out, we have checked the writing grammars in detail throughout the article and revised them. Besides, we have provided more discussion in the introduction to enrich the background research. Then, based on the three amendments you suggested, we thought about it carefully and responded and revised them.

Comment.1: The SLM and LC layer may introduce some noise due to scattered light, or diffraction by the electrode structures. Does it influences the precision of the aberration measurements ?

Response: Thank you for your careful work on this issue. Structural features of LCoS-SLM would cause some noise and scattered light. We could observe optical phenomena such as edge effect and diffracted light. Edge effect leads to irregular wavefront distortion along the four edges of the SLM head, which could be excluded by defined area of interest in the interferometer data. The diffracted light will not on the measurement optical axis of the interferometer so it could not influence the measurement data.

Comment.2: The strategy shown in Fig 2 requires computation time. Which is the time required to calculate and then to display the whole correction on each pixel of the SLM. At which speed to generate a new wave ?

Response: Thank you for your kind consideration on this issue. The whole measurement and computation procedure could take about 13 minutes. The first part time consumption is to upload the two sequences of calibration hologram on the LCoS-SLM and get the phase data by the unwrapping algorithm of the interferometer. Because the software of the interferometer is not open source to us, so each phase map is got by manually operation, which takes about 10 seconds. The time cost of this part is 6 minutes. After these calibration phase data has been got, the rest computation to get the pixel-to-pixel PRM and PCM matrix could be done automatedly by a MATLAB program, which takes about 5minutes. Then the hologram with PCM corrected would take about 90 seconds. As soon as it is uploaded on the LCoS-SLM, a new wavefront could be got. The experiment computing platform is a laptop with i7 CPU and 8G memory.

Comment.3: After correction and no change of the environment, which reproducibility of the X Y addressing electronics on the pixel phase ?

Response: Thank you for such valuable and thorough comments. In our lab with a stable airflow, we used the environment test function of the interferometer software to measure the wavefront aberration within 10seconds. The test was executed several times during 20minutes and the largest wavefront distortion P-V value is about 1/20 λ. So, if there is no change of the environment such as air turbulence and temperature variation, the best of the reproducibility of the phase generated by the LCoS-SLM should be 1/20λ.

We tried our best to improve the manuscript and made some changes in the manuscript. These changes will not influence the main content and framework of the paper and be marked in red in revised paper.

We appreciate for Editors and Reviewers’ warm work earnestly, and hope that the correction will meet with approval.

Once again, thank you very much for your comments and suggestions

Best regards,

Corresponding authors:

Xiaodong Zhang, [email protected]

Round 2

Reviewer 1 Report

This work proposes a phase-compensation method based on the calibration with two reference phase maps and further correction through lookup tables. The authors managed to improve the manuscript following our recommendations and provided satisfactory commentaries to all of my queries. I suggest the authors to double-check the manuscript for correcting possible grammar errors.